# Interactive Responses of *Solanum Dulcamara* to Drought and Insect Feeding are Herbivore Species-Specific

**DOI:** 10.3390/ijms19123845

**Published:** 2018-12-03

**Authors:** Duy Nguyen, Yvonne Poeschl, Tobias Lortzing, Rick Hoogveld, Andreas Gogol-Döring, Simona M. Cristescu, Anke Steppuhn, Celestina Mariani, Ivo Rieu, Nicole M. van Dam

**Affiliations:** 1Molecular Interaction Ecology, Institute of Water and Wetland Research, Radboud University, P.O. Box 9010, 6500 GL Nijmegen, The Netherlands; rickhoogveld1990@gmail.com (R.H.); nicole.vandam@idiv.de (N.M.v.D.); 2Molecular Plant Physiology, Institute for Water and Wetland Research, Radboud University, PO Box 9010, 6500 GL Nijmegen, The Netherlands; c.mariani@science.ru.nl (C.M.); i.rieu@science.ru.nl (I.R.); 3German Centre for Integrative Biodiversity Research (iDiv) Halle-Jena-Leipzig, Deutscher Platz 5e, 04103 Leipzig, Germany; yvonne.poeschl@informatik.uni-halle.de (Y.P.); andreas.gogol-doering@mni.thm.de (A.G.-D.); 4Institute of Computer Science, Martin Luther University Halle-Wittenberg, Von-Seckendorff-Platz 1, 06108 Halle, Germany; 5Molecular Ecology, Institute of Biology, Dahlem Centre of Plant Sciences, Freie Universität Berlin, Albrecht-Thaer-Weg 6, 14195 Berlin, Germany; tobias.lortzing@fu-berlin.de (T.L.); a.steppuhn@fu-berlin.de (A.S.); 6Molecular and Laser Physics, Institute for Molecules and Materials, Radboud University, PO Box 9010, 6500 GL Nijmegen, The Netherlands; s.cristescu@science.ru.nl; 7Institute of Biodiversity, Friedrich Schiller University Jena, Dornburger-Str. 159, 07743 Jena, Germany

**Keywords:** *Solanum dulcamara*, *Spodoptera exigua*, *Leptinotarsa decemlineata*, drought response, plant defence, hormonal signalling, transcriptional regulation

## Abstract

In nature, plants are frequently subjected to multiple biotic and abiotic stresses, resulting in a convergence of adaptive responses. We hypothesised that hormonal signalling regulating defences to different herbivores may interact with drought responses, causing distinct resistance phenotypes. To test this, we studied the hormonal and transcriptomic responses of *Solanum dulcamara* subjected to drought and herbivory by the generalist *Spodoptera exigua* (beet armyworm; BAW) or the specialist *Leptinotarsa decemlineata* (Colorado potato beetle; CPB). Bioassays showed that the performance of BAW, but not CPB, decreased on plants under drought compared to controls. While drought did not alter BAW-induced hormonal responses, it enhanced the CPB-induced accumulation of jasmonic acid and salicylic acid (SA), and suppressed ethylene (ET) emission. Microarray analyses showed that under drought, BAW herbivory enhanced several herbivore-induced responses, including cell-wall remodelling and the metabolism of carbohydrates, lipids, and secondary metabolites. In contrast, CPB herbivory enhanced several photosynthesis-related and pathogen responses in drought-stressed plants. This may divert resources away from defence production and increase leaf nutritive value. In conclusion, while BAW suffers from the drought-enhanced defences, CPB may benefit from the effects of enhanced SA and reduced ET signalling. This suggests that the fine-tuned interaction between the plant and its specialist herbivore is sustained under drought.

## 1. Introduction

In natural environments, plant growth is constrained by a variety of biotic and abiotic stress factors, which often occur in different combinations. Adaptive responses to abiotic stresses, such as drought, and to biotic stresses, such as insect herbivory, are tightly controlled by hormonal signalling. This allows plants to only deploy adaptive responses when needed. The latter is relevant, since adaptive responses may be very costly in terms of resources [1,2,3]. When plants are under concurrent stresses, these signalling pathways interact to regulate plant responses [4,5,6]. However, how this hormonal cross-talk occurs is poorly understood.

Many plant adaptive defence strategies against herbivores have evolved as a consequence of their longstanding evolutionary arms race. The jasmonic acid (JA) signalling pathway mediates many direct defence responses, such as the production of alkaloids, protease inhibitors (PIs), and polyphenol oxidases (PPOs) [7]. Herbivore-induced responses are also influenced by other signalling hormones such as abscisic acid (ABA), ethylene (ET), and salicylic acid (SA). Specific combinations of signalling hormones are triggered upon the recognition of damage patterns—sucking or chewing—combined with cues called herbivore-associated molecular patterns (HAMPs) [8,9,10]. In addition, the responses to specific herbivores and their HAMPs differ between plant species. While *Manduca sexta* herbivory induces JA and ET accumulation in *Nicotiana attenuata* plants, *Spodoptera exigua* herbivory induces JA and SA accumulation [11]. However, herbivory by *S. exigua* on *Arabidopsis thaliana* elicits JA and ET accumulation [12], whereas *Pieris rapae* caterpillars elicit JA and ABA accumulation on the same plant [13]. The simultaneous induction of different hormonal signals results in cross-talk, which eventually results in a specific defence response [8]. JA and ABA co-induce the transcription factor MYC2, which regulates the expression of defence-related genes (e.g., the ones encoding vegetative storage proteins, lipoxigenases, or involved in glucosinolate biosynthesis) and resistance to chewing herbivores [13,14,15,16]. ET signalling also induces several JA-dependent defences, such as the accumulation of nicotine and Mir1-CP [17,18]. By contrast, ET signalling, via ERF1/ORA59 transcription factor activity, antagonises JA/ABA-responsive defence-related genes, thus suppressing plant resistance to several insects [14,19,20]. Similarly, SA signalling induced by insect herbivory, for example by *Frankliniella occidentalis* on *Arabidopsis* and *Leptinotarsa decemlineata* on tomato (*Solanum lycopersicum*), suppresses several JA-dependent defences and increases insect performance [21,22]. Based on these studies, it is now generally accepted that plant–insect interactions are tightly regulated by the species-specific hormonal profiles that are induced upon herbivory.

The co-evolutionary history of the herbivore with a plant species may play a critical role in these species-specific responses [23]. Herbivores specialised on a few closely related host plants would evolve ways to tolerate plant defences. HAMPs of specialist herbivores may suppress defence responses by manipulating their host plant’s hormonal signalling network to their advantage [8,9,10]. In turn, host plants could adopt more effective recognising mechanisms and responses to defend against specialist herbivores [24,25]. On the other hand, due to their broad diet range, generalist herbivores should have more general mechanisms to tolerate a large variety of plant defences, and may be less able to manipulate plant responses. Therefore, it is predicted that specialist and generalist herbivores may induce distinct responses in a particular plant species [26].

Plant responses to drought are also mediated by an interactive hormonal signalling network. ABA plays a central role in the induction of drought responses, including stomatal closure, leaf senescence, and the maintenance of primary root growth [27,28,29]. Under drought, the upregulation of JA and ET signalling induces leaf senescence [30], while JA and SA interact positively with ABA signalling to induce stomatal closure [31,32,33]. ET, contrarily, has a strong antagonistic interaction with ABA in regulating stomatal closure and the development of shoot and root growth under drought stress [34,35,36].

As many of the hormonal pathways activated by herbivores and drought overlap, substantial cross-talk is expected when these stresses occur simultaneously. This likely alters the outcome of the molecular network regulating plant defences, and subsequently plant resistance to insect herbivores. Indeed, drought lowered concentrations of defensive metabolites in *Alliaria petiolata* plants and enhanced the performance of *P. brassicae* caterpillars [37]. Moreover, drought affected the hormone levels and volatile emissions of *Brassica oleracea* challenged by *Mamestra brassicae* herbivory. This resulted in female moths preferring to oviposit on drought-stressed plants [38]. In contrast, drought reduced the performance of the sap-sucking herbivores *Creiis lituratus* and *Myzus persicae* on *Eucalyptus dunnii* and *B. oleracea* var. *capitata*, respectively [39,40]. These contrasting results suggest that the interaction between drought and defence responses is governed by differences in feeding strategies and herbivore-specific signalling.

To unravel the molecular basis of herbivore-specific interactions with drought stress responses, we studied the effect of drought on herbivory by two chewing herbivores: the generalist *S. exigua* (beet armyworm; BAW) and the specialist *L. decemlineata* (Colorado potato beetle; CPB). As a model host plant, we used *Solanum dulcamara*, which is a wild relative of tomato and potato plants that has a broad habitat, ranging from lake boarders to the dry coastal dunes [41]. In addition, it has a rich herbivore community, which includes the specialist CPB [42]. Responses of *S. dulcamara* to drought and flooding have been shown to differentially affect BAW-induced responses [43]. We hypothesized that the plant’s response to drought would have distinct interactions with herbivore-induced responses, resulting in different levels of plant resistance to the specialist and the generalist herbivore. To test this, we conducted bioassays to assess insect performance on well-watered and drought-stressed plants. We combined these with experiments analysing the plant hormonal responses and transcriptomic profiles under the same stress combinations. Our results show that interactions between responses to drought and insect feeding are herbivore species-specific. CPB performed slightly better on drought-stressed plants, whereas BAW performance significantly decreased. Drought enhanced the CPB-induced accumulation of JA and SA and suppressed ET emission. While CPB herbivory on drought-stressed plants enhanced photosynthesis-related and pathogen responses, BAW herbivory further upregulated herbivore-induced transcriptional responses, resulting in the reduced performance of BAW only. This suggests that the specific interaction between the specialist CPB and *S. dulcamara* may be sustained under abiotic stress conditions, to the benefit of the herbivore.

## 2. Results

### 2.1. Drought Treatment Differentially Affects Herbivore Performance

The effect of drought on the insect resistance of *S. dulcamara* was examined by subjecting plants under well-watering (control) and drought treatments to herbivory treatments by either BAW or CPB larvae. Insect weight gain was used as a proxy measure of plant resistance to insect herbivores. The result showed a statistically significant interactive effect on insect weight gain between water treatments and herbivore species (*F*(1, 44) = 6.734, *p* = 0.013). After a five-day feeding period on drought-treated plants, BAW larvae gained significantly less weight (*p* = 0.045), whereas CPB larvae tended to gain more weight (*p* = 0.081) compared with their conspecifics feeding on control plants (Figure 1). Thus, only the performance of BAW was negatively affected when feeding on drought-stressed plants, indicating an increased resistance of *S. dulcamara* plants to BAW, but not to CPB when under drought.

### 2.2. Water Availability Interacts with Herbivory in Regulating Plant Hormone Levels

To investigate the plant hormonal responses underlying changes in herbivore performance on plants under different watering regimes, we measured concentrations of JA, JA-isoleucine (JA-Ile), ABA, SA, and ET in *S. dulcamara* plants under control and drought treatments with or without 48-h herbivory by BAW or CPB. As individual stress factors, herbivory by either BAW or CPB significantly induced the accumulation of JA, JA-Ile, and ABA under control treatment (Figure 2A–C, Appendix A). However, BAW herbivory had significantly stronger effects on the induction of these hormones than CPB, despite the similar damage levels caused by the two herbivory treatments (Appendix A). Moreover, there was a statistically significant interaction between water and herbivory treatments with regard to the accumulation of JA and JA-Ile (Appendix A), mainly because drought only enhanced CPB-induced JA/JA-Ile accumulation (Figure 2A,B). Drought stress alone significantly induced ABA accumulation (Figure 2C), which likely overruled the potential effects of insect herbivory under combined stress conditions (Figure 2C).

Even though neither drought nor herbivory treatments alone affected SA accumulation, CPB herbivory enhanced SA accumulation in drought-treated plants (Figure 2D). However, this effect was not observed for BAW herbivory, resulting in significantly higher SA levels in CPB-damaged plants compared to undamaged and BAW-damaged plants under drought stress.

Ethylene emission of well-watered plants was similarly increased upon 24 h of herbivory by BAW and CPB (Figure 3B). Drought did not affect plant intrinsic or BAW-induced ET emissions (Figure 3A,C). Conversely, CPB-induced ET emissions of drought-treated plants were significantly lower than those of well-watered plants after 24 h of herbivory. This difference was further increased after a 48-h period of herbivory (repeated measures, *F*(1, 10) = 5.728, *p* = 0.038; Figure 3D). This resulted in a significant interaction between water availability and CPB herbivory on ET emission rates (*F*(2.29, 22.90) = 3.600, *p* = 0.038).

In conclusion, water availability interacts with CPB herbivory in regulating plant hormone levels, in which drought enhanced the CPB-induced accumulation of JA, JA-Ile, and SA, and suppressed the herbivory-induced ET emission in *S. dulcamara*. In contrast, the increases in JA, JA-Ile, and ET levels upon BAW herbivory were independent of watering regimes.

### 2.3. Transcriptional Regulation

Differences in plant hormonal responses upon herbivory by generalist or specialist insects may trigger diverging transcriptional responses. To analyse this, leaf RNA samples of the experimental plants used for hormone quantification were subjected to microarray analysis. Of all 33,957 targets, 25,570 had normalized expression values higher than the intensity threshold. These targets were selected for further analyses, and are hereupon referred to as ‘genes’. qPCR analyses on a set of selected genes were used to validate the results of the microarray analyses. This showed a highly significant correlation between genes’ expression levels, which were determined by the two methods (Pearson *R* = 0.913, *p* < 0.001; Appendix A). Multidimensional scaling analysis showed that both insect herbivory treatments significantly impacted the transcriptomic response (Figure 4A). Gene set enrichment analyses (GSEAs) identified 440 significantly affected biological processes (BPs). The clustering of their enrichment scores showed that herbivory by CPB or BAW larvae had similar effects on most BPs (87.5%, or 385 BPs, having at least one comparison with statistically significant normalized enrichment scores (NES; Figure 5). In total, 165 BPs were always significantly induced in the same direction by the two insects, regardless of watering regimes (Figure 4D, part i). The majority of the commonly regulated BPs (145) was upregulated, including many BPs related to responses to insect or JA biosynthesis and signalling (Table 1, Figure 6). Among the 165 genes that were upregulated by both insects regardless of watering regimes (Figure 4C, upregulated genes in part i), many are involved in JA biosynthesis and secondary metabolism (particularly phenylpropanoids) or encode for protease inhibitors (PIs) (Appendix A). Together, these genes represent the common response to insect herbivory in *S. dulcamara*.

Drought treatment had a very different effect on the plant transcriptome than herbivory treatments (Figure 4). Overall, it enhanced the general plant response to both herbivores, as indicated by the 587 additional genes that were induced by both insects (281 upregulated, 306 downregulated; Figure 4C, part f) in drought-stressed plants.

### 2.4. Drought and Insect Herbivory Increase Serine-Type PI (serPI) Levels and PI Gene Expression

As PI activity is a well-known defence against insect herbivores, we also analysed levels of serine PI (serPI), which inhibit proteases with a serine at their active sites, and the regulation of PI-related genes of *S. dulcamara* plants under drought and herbivory treatments. As expected, insect herbivory significantly increased serPI levels of *S. dulcamara* plants (Figure 7A, Appendix A). Moreover, drought induced serPI levels as well as the expression levels of four PI genes (comp460, 251, 255, and 1799; Appendix A). There was no statistically significant interaction between water availability and herbivory treatments on serPI levels. However, when comparing within each watering treatment, herbivory only significantly increased serPI levels in drought-treated plants, and the effect of CPB herbivory on serPI levels was more pronounced than that of BAW. In addition, drought had a significant and positive main effect on the total protein contents of *S. dulcamara* leaves (Figure 7B).

### 2.5. BAW Elicits More Prominent Responses than CPB, Especially under Drought

The above findings on transcriptional regulation and serPI levels, however, did not explain the reduced performance of BAW on *S. dulcamara* plants under drought. We further investigated the transcriptomic data to identify the differential effects of drought on plant responses to BAW and CPB. Since SA signalling is considered a negative regulator of insect-induced JA-dependent defences, we hypothesized that the CPB-induced SA accumulation would suppress plant responses to CPB compared to responses elicited by BAW. The plot of directional changes in gene expression under single and combined treatments (Figure 8) showed that BAW herbivory indeed affected more genes than CPB. This applies both to the numbers of upregulated (‘no < herbivore’, right horizontal) and downregulated (‘no > herbivore’, left horizontal) genes. Correspondingly, 61.5% of the herbivore-upregulated BPs in control plants were induced more strongly by BAW herbivory than by CPB (Figure 5, blue parts of CPB-BAW [control] comparison). This effect was statistically significant for 26 BPs (Appendix A). The difference in response to BAW and CPB was even more prominent on drought-stressed plants (Figure 5, CPB-BAW [drought]), with 62 BPs upregulated significantly stronger by BAW than CPB (Appendix A). Many of these BPs are related to cell wall biogenesis and organization (15 BPs), as well as the metabolism of carbohydrates (eight), lipids (seven), and secondary metabolites (eight). Conversely, CPB herbivory upregulated only one BP (GO:0019216, regulation of lipid metabolism) significantly stronger than BAW in control plants and four unrelated BPs (GO:0006626, 0001510, 0010421, 0015757) in drought-treated plants. Similar patterns were found among the herbivore-downregulated BPs, the majority of which were significantly more strongly affected by BAW than CPB herbivory, particularly those related to photosynthesis (Figure 5 and Appendix A). Together, these results clearly show that BAW had more pronounced effects than CPB on herbivore-mediated responses, especially in drought-stressed plants.

### 2.6. Drought Enhances Specific Responses to Each Herbivore

While drought treatment enhanced the common transcriptional responses to both BAW and CPB, it enhanced the plant’s specific responses to each herbivore as well. Plants under drought treatment induced 569 more genes (309 upregulated, 260 downregulated; Figure 4C, part g) in their specific response to BAW herbivory. When CPB was feeding, 163 unique genes responded (85 upregulated, 78 downregulated; Figure 4C, part c). GO annotation tests on the BAW-specific response in control and/or drought conditions (417 + 285 + 569 genes; Figure 4C, sums of part o + k + g) showed that upregulated genes were primarily involved in JA biosynthesis, cell wall biogenesis and organization, and carbohydrate metabolism, while downregulated genes were mainly involved in photosynthesis (Appendix A). There was no enrichment of responses related to defence against insects in the CPB-specific response (4 + 3 + 163 genes; Figure 4C, sums of part a + b +c). This CPB-specific response contained a large group of 41 genes, which were annotated as ribosomal proteins and/or involved in protein translation (Appendix A). The 10 most upregulated or downregulated genes in the specific responses of drought-treated plants to each herbivore are shown in Appendix A. Notably, five of these CPB-specific upregulated genes were annotated as responsive to pathogens, including *osmotin-like protein*, *patatin-like protein* 3, *alternative oxidase*, *universal stress protein*, and *blue copper protein*. On the other hand, the BAW-specific upregulated genes included two terpene synthases (*Sesquiterpene Synthase1* and *Limonene synthase*).

### 2.7. Herbivore Responses that Were not Induced by CPB in Drought-Stressed Plants

In addition to the insect-specific induced responses, we analysed the part of the common herbivory response that was no longer significantly induced when CPB herbivory occurred on drought-treated plants. We zoomed in on these responses because it may help to understand why CPB performance was not so negatively affected by drought-stressed plants as that of BAW. Forty BPs fell into this category (Figure 4D, part h; listed in Figure 9). Among them, 34 BPs were no longer significantly upregulated by CPB herbivory on drought-treated plants, including processes described as response to insect (GO:0009625, as shown in Figure 6) or related to hormonal signalling (four), cell wall organization and biogenesis (six), lipid (six) and secondary metabolism (four BPs in biosynthesis of suberin, pigments, tetracyclic triterpenoid and sulphur compounds). The other six BPs, which were no longer significantly downregulated by CPB herbivory on drought-treated plants, included three BPs related to photosynthesis and two related to carbohydrate metabolism. At the gene level, 30 significant genes (17 upregulated, 13 downregulated) fitting these expressing patterns were found (Figure 4C, part h; Appendix A). Notably, among the herbivore-responsive genes that were no longer significantly upregulated by CPB on drought-treated plants, five are involved in cell wall remodelling and carbohydrate metabolism (a cellulose synthase-like protein, a pectin acetylesterase-like protein, a glucan endo-1,3-beta-glucosidase, a UDP-d-glucuronate 4-epimerase, and a UDP-glucosyltransferase). These processes and genes might contribute to a better defence of drought-treated *S. dulcamara* plants against BAW, but not to CPB, resulting in the reduced performance of BAW.

### 2.8. Plant Responses to Drought Are More Prominent under CPB Herbivory

Interestingly, the circular histogram (Figure 8) also revealed that drought induced many more genes in the treatment combination with CPB herbivory than with BAW: 116 upregulated (green dots towards ‘control < drought’) and 53 downregulated (green dots towards ‘control > drought’) genes compared to 41 upregulated (red dots towards ‘control < drought’) and seven downregulated (red dots towards ‘control > drought’) genes, respectively. A GO distribution test of the drought-upregulated genes specifically when in combination with CPB (86 genes) showed the enrichment of genes involved in photosynthesis and/or functioning in chloroplasts (Appendix A), including isopentenyl diphosphate biosynthesis (GO:0019288), chlorophyll biosynthesis (GO:0015995), and photosynthetic electron transport chain (GO:0009767). This indicates that plant physiological processes, particularly photosynthesis, may be less constrained by CPB than by BAW herbivory under drought stress.

## 3. Discussion

In this study, we found that the performance of BAW, but not CPB, was reduced when feeding on *S. dulcamara* plants under drought stress. Correspondingly, herbivory by CPB and BAW induced different hormonal and transcriptional responses in *S. dulcamara* under well-watered, control, and drought conditions.

Under control water conditions, BAW herbivory more strongly induced the accumulation of the defence-related hormones JA, JA-Ile, and ABA, as well as related sets of biological processes at the transcriptional level. This is corroborated by a much stronger effect of BAW herbivory on plant transcriptional responses, including many BAW-specific upregulated genes involved in JA biosynthesis, cell wall biogenesis and organization, and carbohydrate metabolism. In *Arabidopsis*, another specialist chewer, *P. rapae*, also induced a weaker transcriptional response than the generalist BAW, which was attributed to the differences in the amount and timing of herbivore-induced ET production [12,24]. However, we did not observe any differences in ET production induced by CPB and BAW in *S. dulcamara*. Instead, our results might be explained by the stronger SA accumulation in CPB-damaged plants. This has been shown to suppress defence responses to insect herbivores via antagonistic cross-talk with the JA signalling pathway [44,45].

### 3.1. Drought Enhances Defence Responses to BAW

The hormonal response to the generalist BAW was not substantially affected by drought. This was possibly due to the strong effect of BAW herbivory alone, which may have maximised the physiological levels of JA and ABA in leaf tissues. Nevertheless, drought enhanced the transcriptional responses to BAW herbivory and the production of serine PIs. BAW herbivory on drought-stressed plants also led to stronger transcriptional responses related to cell wall remodelling and carbohydrate and secondary metabolism. These responses may be important for the acquisition of resources necessary to mount defences against pathogens and insect herbivores [46,47]. Moreover, an increase in cell wall components, such as cellulose and lignin, is strongly linked with the reduced palatability of plant tissues to herbivores [48]. The interaction between responses of *S. dulcamara* to drought and BAW herbivory, which is possibly due to a synergistic effect of elevated ABA levels on JA signalling [43], may maximise the protection of valuable leaf tissues of drought-stressed plants against herbivores. In contrast, a lack of increased ABA accumulation under drought in *Brassica* plants was associated with insect preference for drought-stressed plants over well-watered ones [38]. These insights support the view that drought may promote defence production and plant resistance to some insect herbivores via the synergism of JA and ABA signalling [6].

### 3.2. Interaction between Drought and CPB Herbivory Responses Benefits the Herbivore

In contrast, hormonal responses to the specialist CPB strongly interacted with the drought response. This was illustrated by the increase of herbivore-induced accumulation of JA, JA-Ile, and ABA, the specific SA accumulation, and the suppression of herbivore-induced ET emission when CPB herbivory occurred on drought-stressed plants. At the transcriptional level, drought-stressed plants responded less prominently to CPB than to BAW feeding. This applied especially to genes related to cell wall remodelling and the metabolism of carbohydrates, lipids, and secondary metabolites. We suggest that suppressing these responses may retain, and even improve, the palatability of leaf tissues in drought-stressed plants, and thereby benefit the specialist CPB larvae.

How these transcriptional responses are regulated is still unclear, since CPB herbivory on drought-stressed plants induced levels of JA and ABA to the same extent as BAW herbivory. We suggest that this may be a result of the CPB-specific induction of SA accumulation and the suppression of herbivore-induced ET emission in drought-stressed plants. SA signalling can inhibit several JA-dependent defence responses. Some herbivores are known to be able to exploit this antagonistic interaction via the action of herbivore-specific cues, including HAMPs [9,11,44,45]. Moreover, the induction of SA signalling upon herbivory can also be triggered by viruses or the symbiotic bacteria that insects carry in their oral secretion, which mutually benefits the microbes and their insect hosts. SA accumulation in *Arabidopsis* upon *Frankliniella occidentalis* herbivory is indeed caused by the tomato spotted wilt virus that it carries. This results in a better performance of thrips and allows both, thrips and virus, to spread quickly on their host plants [21]. CPB herbivory on tomato and potato also suppresses the induction by wounding and/or herbivory of several JA-mediated defences, such as *PI* genes and PPO activity, resulting in the enhanced performance of conspecifics [49,50]. In tomato, SA accumulation was shown to be caused by several symbiotic bacteria, including the *Stenotrophomonas*, *Pseudomonas,* and *Enterobacter* species, which are present in CPB oral secretion [22]. These bacteria were also detected in several European CPB populations ([51,52]; A. Grapputo, personal communications); therefore, it is likely that they are also responsible for the enhanced SA accumulation in our study. However, the mechanisms underlying this SA response need to be further investigated. The outcome may be different from what has been described in tomato, because neither JA levels nor serine-PI levels were suppressed in *S. dulcamara.*

In addition, the suppression of herbivory-induced ET emission upon CPB on drought-stressed plants may be relevant, because ET signalling promotes several plant defences against insects [17,18]. How drought may suppress CPB-induced ET emission is an intriguing question. One explanatory factor may be the enhanced accumulation of SA in these plants, because SA signalling may inhibit ET biosynthesis [34,53,54,55].

Furthermore, drought increased the differences between the transcriptional responses of *S. dulcamara* plants to herbivory by CPB and BAW. While defence responses to BAW were further enhanced in drought-stressed plants, the specific response to CPB in drought-stressed plants was not enriched with any typical defences against herbivores. Instead, the expression of several genes annotated as pathogen-responsive were strongly enhanced by CPB herbivory in drought-stressed plants. This is likely related to the pathogen response triggered by the enhanced SA signalling upon CPB herbivory [22]. Whereas the effect of BAW herbivory on drought-stressed plants dominated the plant transcriptional responses, drought-stressed plants subjected to CPB herbivory could also mount an adaptive drought response, which is represented by the enhanced expression of many genes involved in chlorophyll biosynthesis and photosynthetic processes. Rehydration during mild and intermittent drought stress often decreases proteolytic activity and results in photosynthetic recovery [56,57,58,59,60]. Moreover, drought strongly increases serine PI contents in *S. dulcamara*, whose activity sustains the sink status of leaves [61,62]. In addition, a large group of genes encoding ribosomal proteins or proteins involved in gene translation were upregulated by CPB feeding. These responses might divert resources of drought-stressed plants away from producing effective defences against CPB. Alternatively, the enhancement of photosynthesis and other primary compounds may maintain the nutritive value of drought-stressed plants. Both processes might contribute to the better performance of CPB on drought-stressed plants compared to BAW as well as compared to the conspecifics on well-watered plants.

### 3.3. The PI Response is Enhanced by Drought, but Does Not Decrease CPB Performance

Herbivory by either BAW or CPB enhanced the expression of PI genes, but not serine PI concentrations, in *S. dulcamara*. The increased serine PI concentrations were only observed in drought-stressed plants, which suggests that the drought-induced PI response prior to herbivory primes the plants for a faster PI production upon herbivory [43,63]. A candidate causing this interaction is the enhanced ABA signalling in drought-stressed plants. ABA is known to enhance the expression of PI genes, and might overrule the suppressive effect of SA signalling induced by CPB [43,64,65]. Furthermore, the lack of increased ET emissions in the combined drought and CPB herbivory treatment suggests that plants might prioritise defence production via the JA/ABA-dependent pathway, which may explain the higher serine PI contents of drought-stressed plants upon CPB herbivory compared to BAW herbivory. However, since only the performance of BAW was reduced on drought-stressed plants, the increased serine PI contents might not be effective against CPB. Coleopteran species, such as CPB, use cysteine proteases to digest their food. Thus, the digestive proteins of beetles are mostly inhibited by cysteine PIs, which falls within a different class of protease inhibitors [66]. Both CPB and BAW are known to be able to compensate for the inhibited gut proteolytic activity by plant PIs by producing other, insensitive proteases [67,68,69,70]. Nevertheless, specialist herbivores are more adaptive to plant defence responses than generalist herbivores [26,71,72]. Particularly, CPB larval growth was much less affected by the wound/JA-induced PIs in potato compared to BAW [73].

In conclusion, we demonstrated that the responses of *S. dulcamara* plants to drought and insect herbivory interact in a species-specific manner. Under drought stress, *S. dulcamara* plants further enhance defence responses and gain resistance to the generalist BAW. On the other hand, CPB herbivory triggers less pronounced responses compared to BAW, probably owing to its specific induction of SA accumulation and suppression of herbivore-induced ET emission. Moreover, plants under drought divert resources towards adaptive responses to drought and pathogen defence responses that may not be effective against CPB. This suggests that a more specialised interaction between *Solanum* species and the specialist CPB is conserved in *S. dulcamara*, which may benefit the specialist when the plant is under drought stress. Our study provides valuable insights to understand the molecular interactions that shape the relationships between plants and different insect herbivores in their natural environments.

## 4. Materials and Methods

### 4.1. Plant Materials

A seed batch of *S. dulcamara* collected from several plants in Goeree (South Holland, The Netherlands) in 2010 was used in this study [43]. Seeds were stratified at 4 °C in the dark for two weeks, and then sown on potting soil in 11 × 11 × 12 cm (W × L × H) pots covered with a 0.5-cm layer of sand. Plants were kept in an insect-free glasshouse for a 16-h photoperiod, with minimum temperatures set to 20 °C/17 °C (day/night). Greenhouse light was supplemented with sodium lamps (600 W, Philips, Amsterdam, The Netherlands) when natural light fell below 250 µmol m^−2^·s^−1^.

### 4.2. Watering Treatments

Thirty-day-old plants were placed in two netted compartments and randomly assigned to one of two watering regimes: well-watered (control) treatment or moderate drought treatment. Control plants were watered to maintain volumetric soil moisture at 22.0 ± 3.3% (mean ± SD), which was monitored using an ML2x ThetaProbe connected to an HH2 Moisture Meter (Delta-T Devices, Cambridge, United Kingdom). From the onset of the drought treatment, drought-treated plants were supplemented with 50 mL of water daily to induce a gradual decrease of soil moisture. After three days, the soil moisture level of these plants had decreased to 10.4 ± 1.3%. At the same time, the mild wilting of the bottom leaves was observed. After five days, soil moisture had decreased to 9.6 ± 1.9%. The drought-treated plants were not watered during the insect herbivory period.

### 4.3. Herbivory Treatments and Insect Performance

Beet armyworm (*S. exigua*; BAW) larvae were reared on a wheat germ-based diet [74] and kept at 25 °C in a climate cabinet (Snijders Scientific, Tilburg, The Netherlands). Colorado potato beetle (*L. decemlineata*; CPB) larvae were reared on a mixed diet of potato and *S. dulcamara* leaves under greenhouse conditions. Both BAW and CPB larvae were fed on *S. dulcamara* leaves for one day, and starved for two hours before being used in the experiments.

Five days after the initiation of the drought treatment, control and drought-treated plants were subjected to herbivory by CPB or BAW larvae or kept as undamaged controls. For herbivory treatments, one or two young third-instar larvae, depending on their sizes, were confined to the youngest fully expanded leaf of each plant using a white mesh bag to inflict a comparable level of leaf damage. For the undamaged (control) plants, leaves at a similar position were enclosed in mesh bags without insects. After 48 h of herbivory, control and treated leaves (*n* = 12 per treatment, 2 × 3 factorial design) were photographed to measure the damaged areas using Fiji software based on ImageJ v. 1.48d (National Institutes of Health, Bethesda, Maryland, USA) [75]; then, they were flash-frozen in liquid nitrogen, and stored at −80 °C.

To assess insect performance, the larvae that damaged the plants were weighed and moved to younger, undamaged leaves. After another 72 h of feeding, the larvae were weighed again. Total weight gain was calculated by subtracting the initial mass from the final mass and divided by numbers of larvae used per plant to obtain the average weight gain.

### 4.4. Quantification of PI and Total Protein Contents

For molecular analyses (microarrays, quantification of hormones, and PI content), three samples per treatment with the most extreme (lowest and highest) leaf damage were excluded to obtain similar damage levels among herbivory treatments. Total protein extraction was adapted from [76] using four mL of 0.1 M potassium phosphate buffer (pH 7.3, 5% (*w*/*v*) PPVP, 0.83% Triton X-100) per mg leaf material. Total protein content (µg mg^−1^ fresh weight (FW)) was quantified using the Bradford assay [77]. Serine-type PI content (serPI, µg·g^−1^ FW, *n* = nine per treatment) was quantified based on an enzyme activity assay [43,78]. Three technical replicates were taken for each sample. All of the assays were performed in an Infinite^®^ 200 PRO plate reader (Tecan, Männedorf, Switzerland) at room temperature.

### 4.5. Hormone Quantification

Extraction and quantification of ABA, SA, JA, and JA-isoleucine conjugate (JA-Ile) was done according to the protocol modified from Wang et al. [79] using an UPLC-ESI-MS/MS Synapt G2-S HDMS (Waters, Milford, MA, USA; details described by Nguyen et al. [43]). All of the hormone concentrations were calculated over the amount of fresh leaf material used (ng·g^−1^ FW, *n* = nine per treatment). For undamaged leaf samples in which hormone (JA and JA-Ile) concentrations were below the detection limit, random data were generated for the purpose of statistical analysis. This was done conservatively by assuming the limit of quantification (LOQ) of the apparatus and adding to it a random noise generated by the R function “rnorm” (mean = LOQ/2, sd = LOQ/5).

### 4.6. Microarray Analysis

For microarrays, the remaining nine samples per treatment were pooled into three biological replicates such that the levels of damage variation were similar among the pools [43]. Samples from plants without herbivory were randomly assigned to pools. This resulted in 18 pools of leaf materials (*n* = three per treatment group). Total RNA was extracted using the RNeasy Plant Mini Kit (QIAGEN, Venlo, The Netherlands). Genomic DNA in RNA samples was removed by DNase I treatment using the TURBO DNA-free Kit (Ambion, Life Technologies, Carlsbad, CA, USA). The 18 RNA samples were randomly allocated to three customized 8 × 60K Agilent microarrays (Oaklabs, Hennigsdorf, Germany) based on the *S. dulcamara* transcriptome [80,81]. The microarray targets 33,957 potential transcripts. This included 29,091 targets that represent 19,333 *S. dulcamara* contigs in a single direction, 4879 contigs in both directions, and 4866 targets that represent multiple contigs in either one or both directions. Microarray hybridisation, image analysis, background and spatial corrections, probe averaging for targets with multiple probes, and the quantile normalisation of intensity data were performed by Oaklabs. The microarray data are available at the National Center for Biotechnology Information (NCBI) Gene Expression Omnibus, accession number GSE122893.

### 4.7. Quantitative PCR (qPCR) Validation of Microarray Data

To validate microarray data, 18 leaf samples, each represents a pool, were used in qPCR analysis. Total RNA was extracted using an RNeasy Plant Mini Kit (Bio-Rad Laboratories B.V., Veenendaal, The Netherlands) and treated with DNase I (Thermo Fisher Scientific, Waltham, MA, USA). cDNA synthesis were performed using iScript cDNA Synthesis kit (Bio-Rad). Real-time qPCR were carried out in a CFX96 Real-Time System (Bio-Rad) using iQ SYBR Green Supermix (Bio-Rad) containing 200 nM of each primer and a two-step standard protocol (45 cycles of 15 s at 95 °C and 30 s at 60 °C) with melt curve analysis. Primer sequences are provided in Appendix A. Quantification of differential expression was performed in according to Rieu and Powers [82].

### 4.8. Gene Set Enrichment Analysis and Clustering of Transcriptomic Response

Gene ontology enrichment in biological processes (BPs) for microarray targets was performed using a previously described annotation [43] using GSEA v. 2.1.0 (Broad Institute, Cambridge, MA, USA) [83]. A cut-off at 0.1 for the false discovery rate (FDR) *q* value was used to determine significantly affected BPs. Cytoscape v. 2.8.3 [84] was used to visualise the enrichment maps. Related clusters and single BPs were grouped and labelled. Hierarchical clustering (Euclidean distance, average linkage) by GENE-E (Broad Institute, http://www.broadinstitute.org/cancer/software/GENE-E/index.html) was used to cluster transcriptomic responses based on normalized enrichment scores (NES) of the gene sets from the enrichment analyses. Only gene sets with at least one significant FDR *q* value from all of the GSEAs were included in the clustering.

### 4.9. On-Line Detection of ET Emission

Separate experiments were set up to measure ET emissions of *S. dulcamara* plants upon drought stress and insect herbivory using a laser-based ET detector in the stop-and-flow mode (ETD-300, Sensor Sense B.V., Nijmegen, The Netherlands) [43]. The youngest fully expanded leaf of each plant was inserted into a customised cuvette. Three cuvettes with leaves subjected to different treatments were measured in parallel. An empty cuvette was always included in each set of measurements to serve as an air reference. First, we assessed ET emissions induced by CPB or BAW on well-watered plants during a 24-h period. After measuring baseline ET emission rates (nL·h^−1^) without herbivory for 20 min at a three L h^−1^ air flow rate, a fourth-instar CPB (*n* = 6) or BAW larva (*n* = eight) was inserted in a cuvette, where they started to feed. Plants without herbivores in the cuvette were kept as controls (*n* = six) and measured in parallel. In a second experiment, we tested the effects the drought treatment on ET emission by undamaged and herbivore-infested plants. After the first two measurements without insect herbivory, one BAW larva was inserted into the cuvette of control (*n* = four) or drought-treated plants (*n* = six) to feed for 24 h. A similar design was used to examine the effect of drought on ET emission during CPB herbivory (*n* = six per watering treatment). These measurements during CPB herbivory were extended to cover a 48-h period. One extra plant under well-watered treatment without herbivory was included to represent the control ET emission rate of undamaged leaves.

### 4.10. Statistical Analyses

Microarray data analyses and statistical tests were done on log_2_-transformed intensity data in R environment (R Development Core Team 2013). Targets with intensity values lower than the intensity threshold determined by density plots (<2) were considered as background noise and discarded from the dataset. Multidimensional scaling (MDS) was performed on a “1-cor”-distance matrix of the dataset without the noise targets. We fitted two-way ANOVA models using the function *lm()* with watering and herbivory treatments as the two factors. ANOVA tables for these models were computed by *anova()*, followed by pairwise comparisons with post-hoc Tukey HSD tests using *TukeyHSD()* and calculation of the false discovery rate (FDR) corrected *p* values using the Benjamini–Hochberg procedure (Benjamini and Hochberg 1995). Cut-offs for targets with significant changes in expression were set at two for fold changes and 0.01 for FDR *p* values.

Data on insect performance, hormone concentrations, and total protein and serPI contents were tested by standard statistics using PASW Statistics v.21 (IBM, Armonk, NY, USA). Levene’s and Shapiro–Wilk tests were performed to check for homogeneity of variance and normality, respectively. Two-way ANOVAs were used for testing the effects and interaction of water availability and 48-h herbivory treatments on hormone concentrations, protein, and PI contents. Hormone data were log_10_-transformed before analysis. These tests were followed by one-way ANOVAs and independent sample Student’s *t*-tests to determine the significant effects of herbivory under each watering treatment, and the effects of watering treatments (with or without herbivory), respectively. All of the significant one-way ANOVAs were followed by Fisher’s least significant difference (LSD) tests. For ET emission rates, repeated measures analyses were used to test for herbivory effects. These analyses also included tests of the between-subjects effects of watering treatments and the interactions with herbivory treatments.

## Figures and Tables

**Figure 1 ijms-19-03845-f001:**
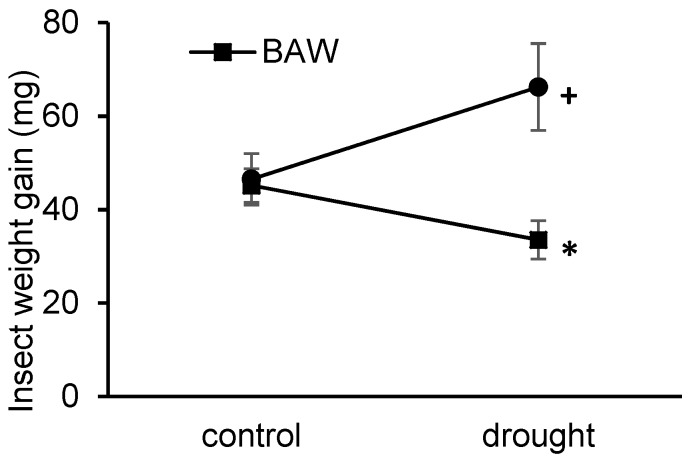
Effects of water availability on insect performance on *Solanum dulcamara* plants. *Spodoptera exigua* (beet armyworm, or BAW; squares) and *Leptinotarsa decemlineata* (Colorado potato beetle, or CPB; dots) larvae were fed for five days on well-watered (control) or drought-treated (drought) plants. + *p* < 0.10 and * *p* < 0.05 obtained from Student’s t tests for the effect of watering treatments on CPB and BAW weight gain, respectively. Error bars are standard errors, *n* = 12 per treatment combination.

**Figure 2 ijms-19-03845-f002:**
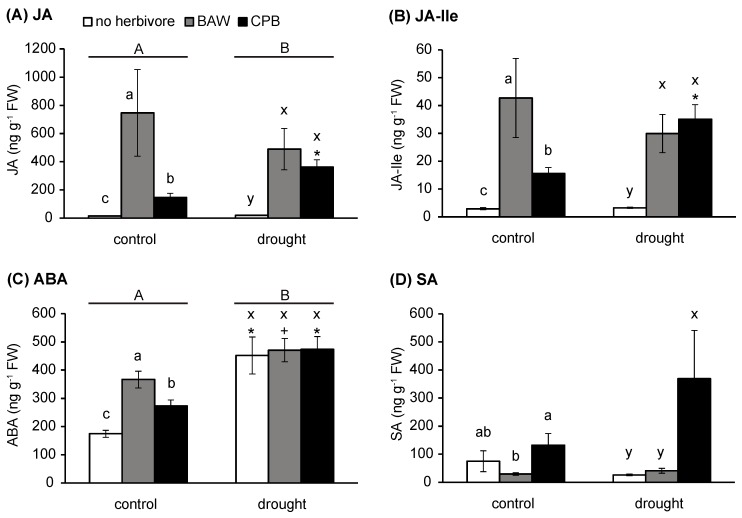
Effects of water availability and insect herbivory treatments on hormone concentrations in *Solanum dulcamara* leaves. (**A**) Jasmonic acid (JA), (**B**) JA-isoleucine (JA-Ile), (**C**) abscisic acid (ABA), and (**D**) salicylic acid (SA). Error bars are standard errors, *n* = nine per treatment combination. Bars indicate values of plants without herbivores (control, white bars), *Spodoptera exigua* (BAW, grey bars) and *Leptinotarsa decemlineata* (CPB, black bars). Different capital letters (A or B) indicate significant main effects (univariate, *p* < 0.05) of watering treatments. Different small letters (a, b, c in control watering regime; or x, y, z in drought treatment) indicate significant differences (least significant difference, or LSD *p* < 0.05) between herbivory treatments within each watering treatment + *p* < 0.10; * *p* < 0.05 obtained from Student’s *t*-tests comparing well-watering (control) and drought treatments for each herbivory treatment. FW: fresh weight.

**Figure 3 ijms-19-03845-f003:**
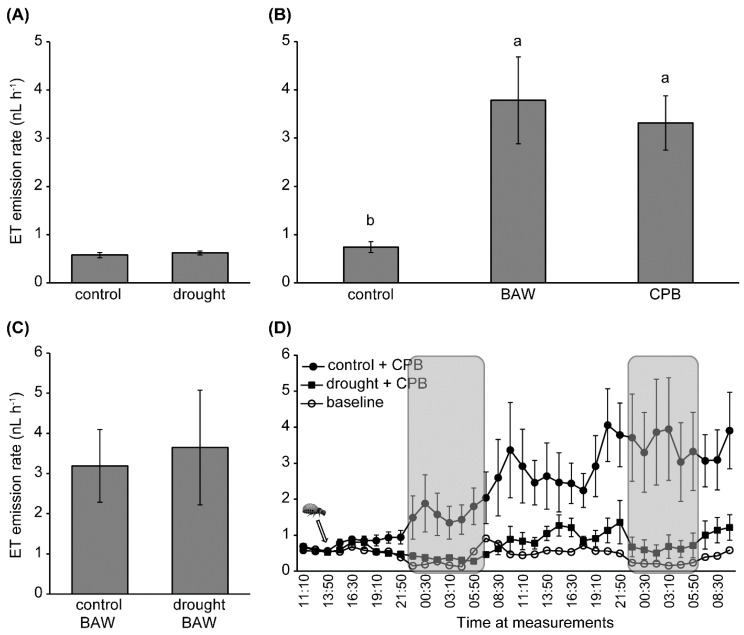
Effects of water availability and insect herbivory treatments on ethylene (ET) emission (nL h^−1^) from *S. dulcamara* leaves. (**A**) Plants were well-watered (control) or under drought treatment for five days before the ET measurements. Each replicate (*n* = seven per treatment) is the average of two consecutive measurements (20 minutes per measurement). (**B**) Leaves that were undamaged (control, *n* = six) or fed on by *Spodoptera exigua* (BAW, *n* = nine) and *Leptinotarsa decemlineata* (CPB, *n* = six) larvae for a 24-h period. Each replicate represents the average emission rate over the whole 24-h period. Error bars are standard errors. Different letters (a, b, c) indicate significant differences between herbivory treatments (LSD *p* < 0.05). (**C**) Plants were under five-day control (*n* = four) or drought (*n* = six) treatments before a 24-h herbivory treatment by BAW. Each replicate depicts the average emission rate over the whole period. (**D**) Plants were under five-day control (filled dots) or drought treatments (filled squares, *n* = six per treatment) before a 48-h herbivory treatment by CPB. The arrow indicates when herbivory treatment started by adding a larva to each cuvette. One extra plant under control treatment without herbivory was included as the baseline of the ET emission rate (empty dots). Shaded blocks indicate dark periods.

**Figure 4 ijms-19-03845-f004:**
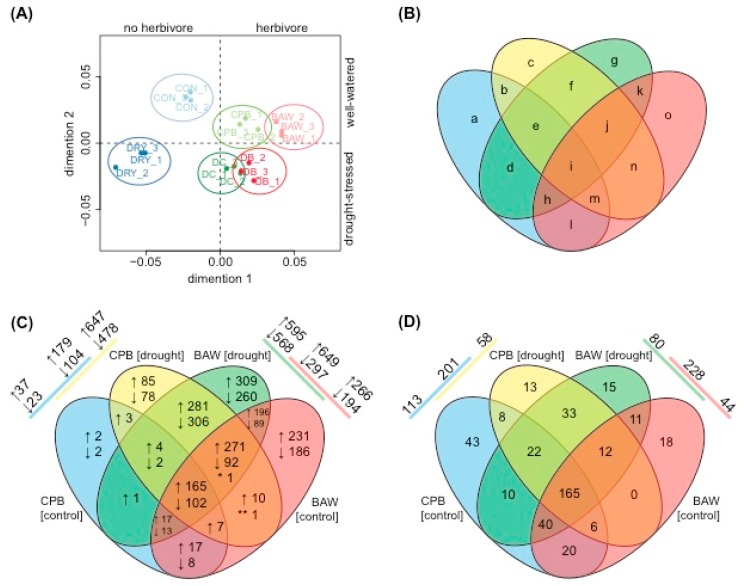
Transcriptomic regulation of *Solanum dulcamara* under combinations of watering and insect herbivory treatments. (**A**) Multidimensional scaling analysis of microarray data of plants under well-watering conditions without (CON) or with herbivory by *Spodoptera exigua* (BAW) or *Leptinotarsa decemlineata* (CPB), or of plants subjected to a moderate drought treatment without (DRY) or with herbivory by BAW (DB) or CPB (DC). Replicates of the same treatments are similarly colour-coded and circled. (**B**) Simplified scheme to clarify parts of Venn diagrams in (**C**,**D**). (**C**) Numbers of upregulated (↑) or downregulated (↓) genes of plants subjected to herbivory by *S. exigua* or *L. decemlineata* compared to undamaged plants under well-watering conditions (BAW [control] and CPB [control], respectively) or drought treatment (BAW [drought] and CPB [drought], respectively). * ↑ in BAW, ↓ in BAW [drought], and CPB [drought]; ** ↑ in BAW, ↓ in CPB [drought]. (**D**) Numbers of induced biological processes (BPs; both upregulated and downregulated) of plants under the same treatments. In the top left and right corners of Venn diagrams in (**C**,**D**) are the total numbers of specific or shared genes or BPs, respectively, induced by each insect on well-watered or drought-treated plants.

**Figure 5 ijms-19-03845-f005:**
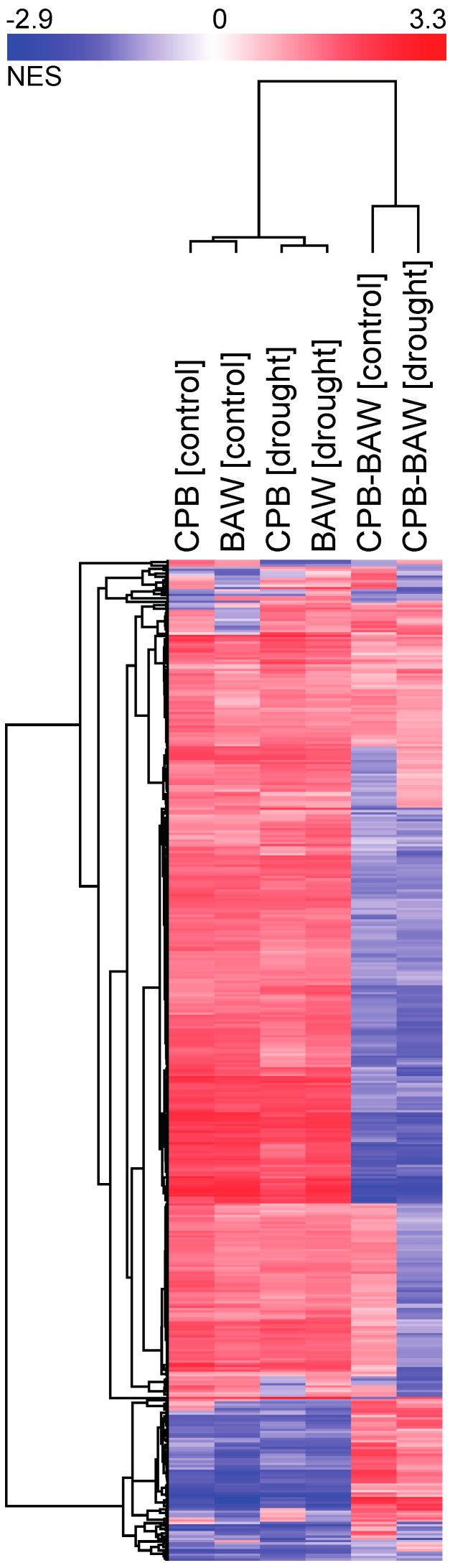
Hierarchical clustering of the normalized enrichment scores (NES) of biological processes (BPs) responding to insect herbivory in *Solanum dulcamara* plants. Each column is a pairwise comparison between two conditions to show the effects of herbivory by *Spodoptera exigua* or *Leptinotarsa decemlineata* under well-watering conditions (BAW (control) and CPB (control), respectively) or drought treatment (BAW (drought) and CPB (drought), respectively), or to show differences between herbivory treatments by the two insects on well-watered (CPB-BAW (control)) or drought-stressed plants (CPB-BAW (drought)). Colours indicate directions of enrichment: red for upregulation (positive NES) and blue for downregulation (negative NES).

**Figure 6 ijms-19-03845-f006:**
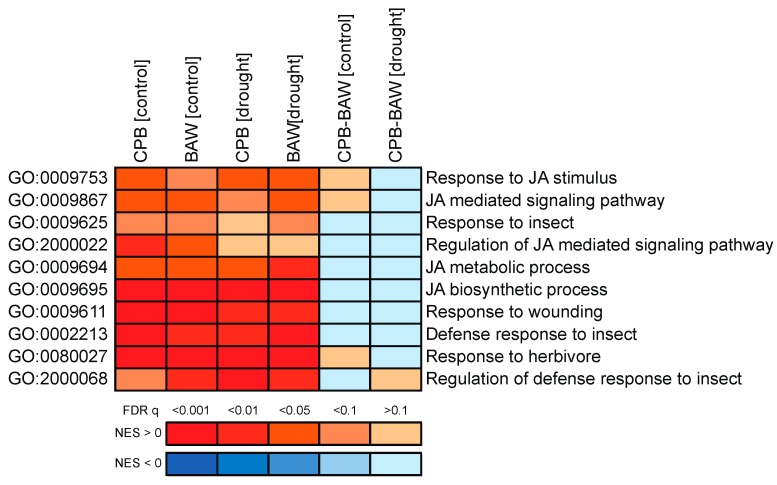
Induction of jasmonic acid (JA)/herbivore-related biological processes (BPs) by insect herbivory in *Solanum dulcamara* plants. Each column is a pairwise comparison between two conditions to show effects of herbivory by *Spodoptera exigua* or *Leptinotarsa decemlineata* under well-watering conditions (BAW (control) and CPB (control), respectively) or drought treatment (BAW (drought) and CPB (drought), respectively), or to show differences between herbivory treatments by the two insects on well-watered (CPB-BAW (control)) or drought-stressed plants (CPB-BAW (drought)). Colours indicate directions of BP enrichment based on normalized enrichment scores (NES) generated by gene set enrichment analyses: red for upregulation (positive NES) and blue for downregulation (negative NES), higher absolute values of NES indicate more significant differences. Colour scales indicate levels of significance based on the associated FDR *q* values. Cut-off for significance is set at *q* = 0.1. GO: gene ontology.

**Figure 7 ijms-19-03845-f007:**
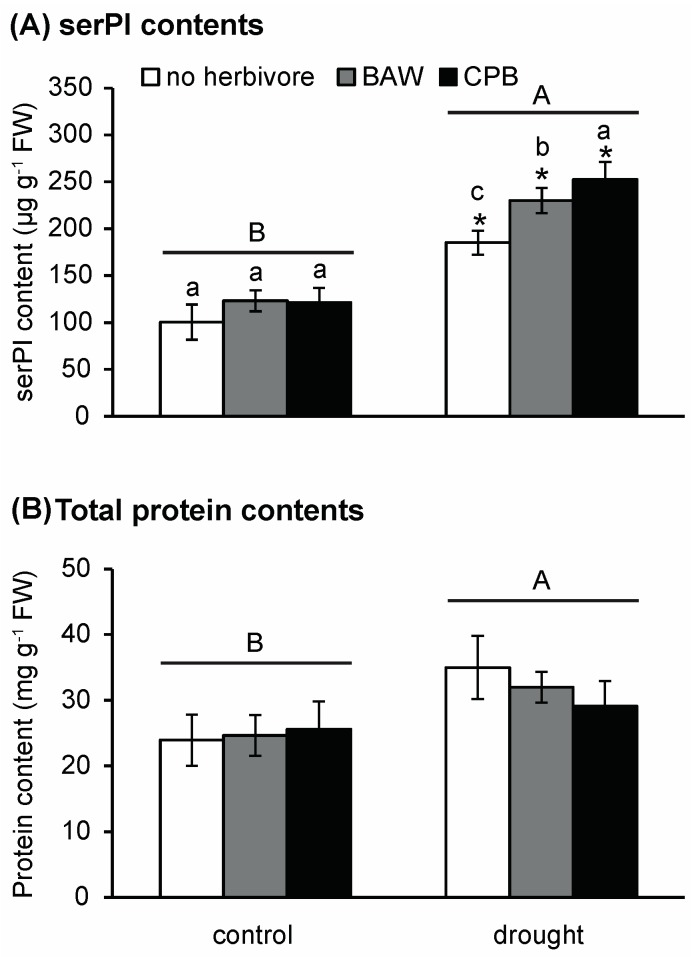
Effects of water availability and insect herbivory treatments on serine-type protease inhibitor (**A**) and leaf total protein (**B**) concentrations of *Solanum dulcamara* leaves. Plants were well-watered (control) or under drought treatment (drought) without herbivory (no herbivore, empty bars) or fed by *Spodoptera exigua* (BAW, grey bars) or *Leptinotarsa decemlineata* (CPB, black bars). Different capital letters (A or B) indicate the significant main effects of watering treatments (*p* < 0.05). Different small letters (a, b, c) within each watering treatment indicate significant differences between herbivory treatments. * *p* < 0.05 obtained from Student’s *t*-tests between well-watering and drought treatments for each insect herbivory group. Error bars are standard errors, *n* = nine per treatment combination. serPI, serine-type protease inhibitor.

**Figure 8 ijms-19-03845-f008:**
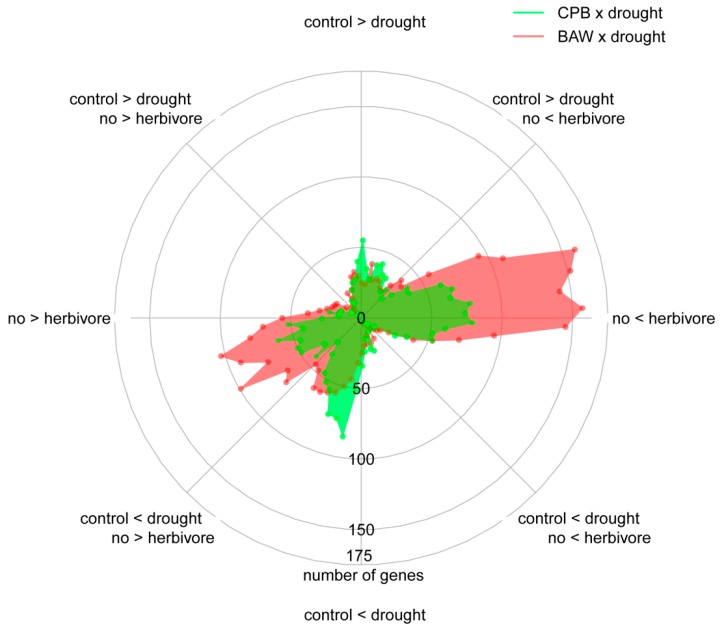
Effects of water availability and insect herbivory treatments on gradient directions of *Solanum dulcamara* transcriptional regulation. Data were generated from a 2 × 2 factorial design of watering treatments (well-watering (control) or drought] and herbivory (with or without) by *Spodoptera exigua* (BAW × drought; red) or *Leptinotarsa decemlineata* (CPB × drought; green), respectively. Each dot indicates the number of genes in one of the 72 categories of the circular histogram (based on gradient direction). Only genes with at least one significant expression change were included in the analysis.

**Figure 9 ijms-19-03845-f009:**
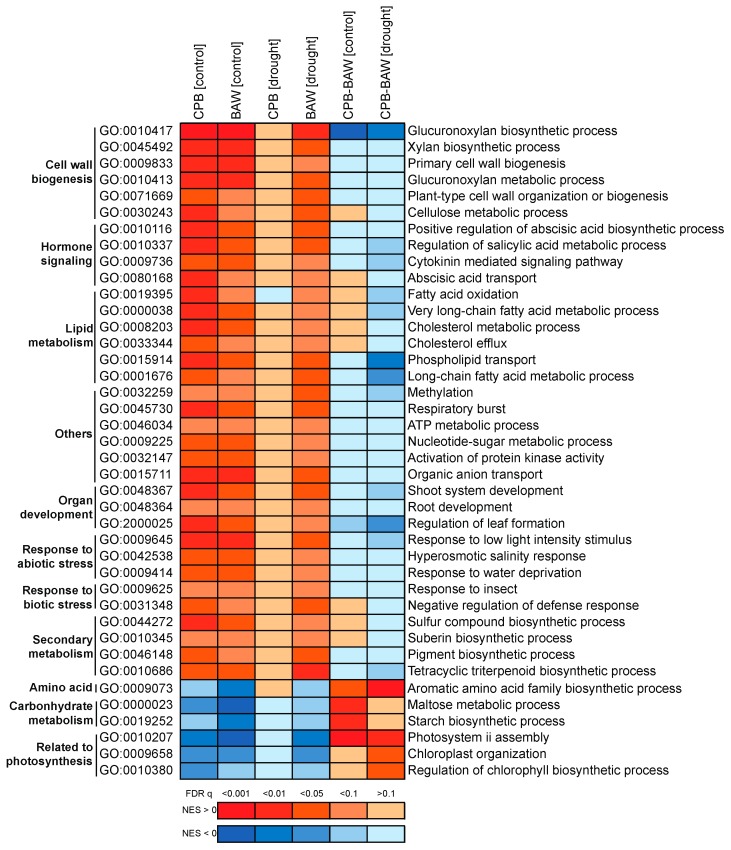
Herbivore-induced biological processes in *Solanum dulcamara* that were no longer significantly induced by *Leptinotarsa decemlineata* (CPB) feeding on drought-stressed plants. Each column is a pairwise comparison between two conditions to show the effects of herbivory by *Spodoptera exigua* or *L. decemlineata* under well-watering conditions (BAW (control) and CPB (control), respectively) or drought treatment (BAW (drought) and CPB (drought), respectively) or to show differences between herbivory treatments by the two insects on well-watered (CPB-BAW (control)) or drought-stressed plants (CPB-BAW (drought)). Colours indicate directions of the BP enrichment based on normalized enrichment scores (NES) generated by gene set enrichment analyses: red for upregulation (positive NES) and blue for downregulation (negative NES); higher absolute values of NES indicate more significant differences. Colour scales indicate levels of significance based on the associated false discovery rate (FDR) q values. Cut-off for significance is set at *q* = 0.1. GO: gene ontology number. GOs of the same class were grouped.

**Table 1 ijms-19-03845-t001:** Classes of biological processes in *Solanum dulcamara* that significantly responded to herbivory by both *Spodoptera exigua* and *Leptinotarsa decemlineata*, regardless of watering regimes.

Group of Biological Processes	Upregulated	Downregulated	Total
Amino acid metabolism	16	0	16
Carbohydrate metabolism	14	0	14
Cell wall remodelling	19	0	19
Developmental processes	4	0	4
Hormonal homeostasis and signalling	14	0	14
Ion homeostasis	7	0	7
Lipid metabolism	11	0	11
Photosynthesis-related processes	1	7	8
Redox homeostasis	8	3	11
Responses to abiotic stresses	6	4	10
Responses to biotic stresses	15	0	15
Secondary metabolism	20	0	20
Other molecular physiological processes	10	6	16
Total	145	20	165

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
