# Peer review of "Interactive Responses of Solanum Dulcamara to Drought and Insect Feeding are Herbivore Species-Specific"

_ijms, 2018, doi:10.3390/ijms19123845_

Round 1
Reviewer 1 Report
This manuscript describes a simple, yet elegant set of experiments designed to investigate the cross-talk of hormones, genes, and specific defense-related proteins after insect herbivory of either a specialist and generalist chewer on Solanum dulcamara under drought or well-watered conditions. Insect herbivory is a major plant stress but often plant experience difference and multiple stresses when growing in natural or agricultural environments. Experiments such as this help elucidate the many biological pathways in plants that allow them to respond differentially to different herbivores.
Overall, the paper is well-written and the scientific methods are sound. My comments are limited but I have a few questions that I would like the authors to address.
1. Line 26: Write out Beet Army Worm and Colorado Potato Beetle in the abstract before using acronyms.
Answer: The common names are now inserted.
2. Line 28: You claim that "Bioassays showed that plants under drought became more resistant to BAW but not CPB". Unfortunately, you can not state this. You did not measure whether or not the plants were more resistant. This would require doing more intestine image analysis, which you did not do. You would have to demonstrate that plants were LESS EATEN by one insect or another under drought conditions vs. well watered conditions. Your study only looked at insect weight, which would show insect tolerance on plants under different watering regimes. According to Supplementary Figure 1, you did measure plant damage, but there are no statistical analyses on the graph or described in the caption that suggest that drought plants with BAW were eaten less than those with CPBs.
Answer: In the literature, both damage and insect performance are used as measures of plant resistance. In most cases this is justifiable as body mass increase of the insect is directly related to how much it consumed, or the amount of damage. However, as we did not explicitly measure this, we cannot be 100% sure. Therefore, we rewrote the sentence on line 28 to better describe the results.
Supplementary Figure 1 shows plant damage after 48h of herbivory, which was too short to result in significant differences in damage between treatments. The results of insect performance in Figure 1 were obtained after 120h of feeding.
3. Lines 125-126. Again you mention plant resistance...per my comment above, I think your manuscript is sound, but you need to either delete or rewrite these statements to reflect insect tolerance/performance rather than plant resistance.
Answer: The sentence is rewritten to better reflect the results as suggested by both Reviewers.
4. Line 158 (and others in figure captions...266)- Your captions in figures 2 and 7 state that you are using a student's t-test to do pair-wise comparisons of control vs. drought treatments. In your methods you describe that you use one-way ANOVA with LSDs for statistics analysis. Can you clarify which we're used. I am concerned by Figures 2 and 7 with 6 treatments (not just a 2X3 factorial design). Wouldn't showing your ANOVA variables with either Bonferroni or post hoc Tukey tests so that your letters represent significant differences across all 6 treatments be more robust?
Q: What was the reasoning for using Fisher's LSD? (i.e. small sample size?)
Answer: As described in Materials and Methods, after 2-way ANOVAs for 2x3 factorial design and when significant main effects were found, ANOVAs (with LSDs) were used to test the effects of insect herbivory on the measurements under each watering treatments, and t-tests were used to test the effects of watering treatments on the measurements for each herbivory conditions (undamaged, BAW- CPB-fed). For the suggestion of the Reviewer, it may not be most statistically appropriate to perform ANOVAs with Tukey for all 6 treatments since in that way the 6 treatments are considered statistically independent, which is not the case of 2x3 factorial design.
We used LSDs instead of more conservative methods, such as Tukey, since the numbers of pair-wise comparisons were limited (maximum 3). In any case, LSDs were considered only when the preceding ANOVAs showed statistically significant effects. This is now mentioned in the M&M section.
5. Line 195: Spell out GSEA
Answer: corrected.
6. Line 245 Define NES briefly. This will allow the reader to better understand how the data were determined to be significantly up or down regulated compared to controls. Was there a cut off? Also enrichment is misspelled on line 245.
Answer: We added more details here (which are also provided in Materials and Methods) as to make the analyses easier to understand when reading the results.
6. Table 1: I know that you are trying to keep things simple, but is there any way to show this by insect rather than lumping both insects into 1 up and 1 down column?
Answer: Here we focus on the biological processes that responded in the same direction upon herbivory by CPB and BAW, hence they are in the same column. Later we focus on the contrasts between the insect. We indeed intended to keep the oversight in the large dataset rather simple in this way.
7. Line 248: Write out Serine Proteinase Inhibitors and give a very brief description of what they are. You allude to cysteine PIs in CPBs in your discussions so a brief (1 sentence) comment on why other PIs (like serPI) might be ineffective would be helpful for a broader audience.
Answer: A short description for serPI is added. As discussed in the Discussion, the plants’ serPI might not be effective against CPB, because CPB depend on cysteine proteases to digest their food.
8. Figure 7: Why was FW used for total protein content and not Dry weight? Could it be possible that lower water content might have accounted for slightly higher protein: FW ratios?
Answer: FW was used since our PI extraction method is based on fresh leaf material. Changes in water content of leaf materials indeed might contribute to changes in protein contents. However, our preliminary experiments (data not included) showed that at the position of the first fully expanded leaf, which were harvested, differences in water contents between leaves of control and drought-stressed plants were very small and would not influence the conclusions we draw.
9. Line 322: I know it is often used in the literature, but using the word "performed" to describe insect weight gain is over-speculative.
Answer: sentence is rewritten.
10. Line 335-336: Again, reword or delete as you did not measure resistance in plants.
Answer: sentence is rewritten.
11. Line 520: This is more of a curiosity question about your methods that needs some clarification. You had 12 plants per treatment and 6 treatments (drought-no herb, BAW, CPB and watered-no herb, BAW, CPB).
Q: Were the insects put on only one leaf for 48 hours? Was there enough leaf material left to get adequate material for hormone and protein analysis given that the remaining 9 leaves from each treatment group/plant were used for RNA isolation?
Answer: the insects were indeed allowed to feed for 48h. However, they were early instars, therefore the range of leaf damage was only 5-25% of the total leaf area. Hormone and protein extractions only required 300mg fresh materials in total. For RNA isolation, about 100mg fresh materials was used. We also pooled 3 plants together to obtain 3 biological replicates per treatment. Consequently we obtained sufficient materials for the extractions.
Q: Were the insects put on only one leaf for 48 hours? Was there enough leaf material left to get adequate material for hormone and protein analysis given that the remaining 9 leaves from each treatment group/plant were used for RNA isolation?
Author Response
Dear Reviewer,
Thank you very much for your evaluation of the manuscript and comments to improve it. We recognise your main concern of the use of the word "resistance" in the manuscript therefore added text as well as rewrote several sentences to better reflect the results of our experiments. More details are added to make the experimental and statistical methods easier to understand as your suggestions. Please find the attached file for details.
Best regards,
Duy Nguyen

Reviewer 2 Report
The manuscript of Nguyen et al. (ijms-388716-peer-review-v1) investigates the effect of drought on plant performance, hormonal and transcriptional regulation of Solanum dulcamara under insect herbivory. Two insect species from different orders with different levels of diet specialisation were chosen: the generalist lepidopteran Spodoptera exigua (BAW) and the specialist Leptinotarsa decemlineata (CBP). Using bioassays with S. dulcamara plants under drought weight gain of CPB was higher than BAW. Chemical analysis showed that only CPB enhanced accumulation of jasmonic acid and salicylic acid, but suppressed ethylene emission, whereas BAW did not so. Using microarray analyses authors show that under drought, BAW feeding enhanced transcription of genes involved in cell-wall remodeling and metabolism of carbohydrates, lipids and secondary metabolites. In contrast, CPB herbivory enhanced several photosynthesis-related and pathogen responses in drought-stressed plants. Authors suggest that this may increase the tissue’s nutritive values by inhibiting production of effective defences. In conclusion, this study indicates that BAW suffers from drought-enhanced defences, whereas CPB benefit from effects triggered by salicylic acid and ethylene signaling. Thus, responses of S. dulcamara to drought and insect herbivory seem species-specific, which in case of CPB, seem to maintain the versatile interaction with its host plant under drought stress.
The manuscript is well-written, most experiments are properly designed, the methods are clearly described and the outcomes of the study enhance our molecular understanding of plant responses towards herbivory under osmotic stress. However, two main issues in the manuscript require careful consideration and explanation by the authors:
1. When testing for differences in induced plant defense between a specialist and generalist herbivore, an optimal comparison should be within the same feeding guild (but here leaf-chewing for CBP versus leaf-snipping for BAW) and also within one phylogenetic lineage (but here Coleopteran vs lepidopteran species). See Ali & Agrawal 2013, Trends Plant Sci. 693, 293–302 for details (also cited by the authors, but in another context). Therefore, it is unclear why not a generalist Coleopteran and/or specialist lepidopteran species was used in addition to CPB and BAW, which would have led to clearer conclusions according to the above issue.
Answer: We are primarily interested in the plant’s response to CPB due to its specific interaction with Solanum species and the fact that it is an important pest to potato plants. As indicated by the Reviewer, it would be ideal to have a generalist Coleopteran species being able to inflict similar levels of leaf damage as CPB larvae for direct comparison. However, it was not possible for us to identify such species – in Dutch populations the plants are commonly colonized by specialist beetles (see Calf and van Dam, 2012). Therefore, we used S. exigua, which is a widely used reference species.
2. The use of the word plant “resistance” and “resistant” throughout the MS is not justified, since herbivory experiments were apparently only carried out by measuring insect weight gain. However, more important from a plant’s perspective when referring to its herbivore resistance is to measure the consumed leaf area. That means that although weight gain differs between two insect species, the consumed leaf area might not necessarily differ between the two species, especially when comparing species from different orders.
Answer: Indeed the consumed leaf area would give a more direct indication of plant resistance to insect herbivores. Insect performance (weight gain, generation time…) is frequently used as proxy measurement for plant resistance. Considering the concern of both Reviewers, we added a sentence to indicate weight gain parameter was used as a proxy of resistance (line 125). We also rewrote concerning sentences to better reflect our data, insect performance rather than plant resistance.
Minor:
- avoid “On the other hand” if you do not add “on one hand” in the sentence before (e.g. line 65, 77, ...)
Answer: As implied by the Reviewer, “on the other hand” can be used with or without the use of “on one hand” in the sentence before. However, we replaced it with other adverbs in several occasions to avoid overusing it.
Author Response
Dear Reviewer,
Thank you very much for your evaluation of our manuscript and the comments to improve it. To address your main concern of the use of the word "resistance", which is shared by the second reviewer, we added text and rewrote some sentences to clarify it as well as to reflect better the data obtained from our experiment. Please see the attached file for other comments.
Best regards,
Duy Nguyen
